# Tracing Training Progress: Dynamic Influence Based Selection for Active Learning

## ABSTRACT

Active learning (AL) aims to select highly informative data points from an unlabeled dataset for annotation, mitigating the need for extensive human labeling effort. However, classical AL methods heavily rely on human expertise to design the sampling strategy, inducing limited scalability and generalizability. Many efforts have sought to address this limitation by directly connecting sample selection with model performance improvement, typically through influence function. Nevertheless, these approaches often ignore the dynamic nature of model behavior during training optimization, despite empirical evidence highlights the importance of dynamic influence to track the sample contribution. This oversight can lead to suboptimal selection, hindering the generalizability of model. In this study, we explore the dynamic influence based data selection strategy by tracing the impact of unlabeled instances on model performance throughout the training process. Our theoretical analyses suggest that selecting samples with higher projected gradients along the accumulated optimization direction at each checkpoint leads to improved performance. Furthermore, to capture a wider range of training dynamics without incurring excessive computational or memory costs, we introduce an additional dynamic loss term designed to encapsulate more generalized training progress information. These insights are integrated into a universal and task-agnostic AL framework termed Dynamic Influence Scoring for Active Learning (DISAL). Comprehensive experiments across various tasks have demonstrated that DISAL significantly surpasses existing state-of-the-art AL methods, demonstrating its ability to facilitate more efficient and effective learning in different domains.

## CCS CONCEPTS

• **Computing methodologies → Learning paradigms**; **Machine learning approaches**.

## KEYWORDS

Active Learning, Dynamic Influence Estimation, Training Dynamics

## 1 INTRODUCTION

Active learning (AL) is a machine learning paradigm that focuses on selecting more informative data points from an unlabeled dataset for annotation [24]. By identifying an efficient training data subset, AL has demonstrated to be an effective method for alleviating the

*ACM MM, 2024, Melbourne, Australia*
© 2024 Copyright held by the owner/author(s). Publication rights licensed to ACM.
ACM ISBN 978-x-xxxx-xxxx-x/YY/MM
https://doi.org/10.1145/nnnnnnn.nnnnnnn

**Unpublished working draft. Not for distribution.**

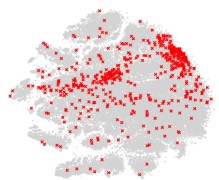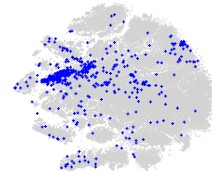

**Figure 1: The red and blue points represent the data samples selected at Epoch 10 and 100, while employing the same AL method with 10% selecting rate. Subsets selected from different checkpoints tend to exhibit significant fluctuation.**

bottleneck challenge of data annotation [1, 20], enabling faster iteration in model development. Since current deep learning models are data-hungry [18], particularly in the context of large foundation models, where the quality of labeled data also significantly impacts model performance [3, 40], actively selecting data has gained increasing attention in recent years [33].

Existing AL approaches can be generally divided into two main categories: uncertainty-based [9, 31] and feature distribution-based [2, 27]. The former utilizes uncertainty estimation to identify hard-to-learn samples, while the latter aims to understand the underlying feature distribution and identify representative and diverse samples. Additionally, a variety of hybrid strategies have been proposed to combine the above these two criteria for AL [4, 36, 39]. While they define the state-of-the-art baselines, the design of these AL strategies are mainly based on human experience, lacking interpretability. Moreover, they are often tailored for specific tasks or labeling budget [4, 11].

To better explain how each selected instance impacts the model performance, gradient-based influence function [16] has been introduced in AL to directly make a direct quantitative evaluation [21, 35]. For example, Liu et al. [21] estimate its expected gradient to calculate unlabeled samples' impact on model performance. Similarly, Wang et al. [35] employ two alternative schemes to calculate the gradient norm, leading to a lower upper-bound of the test loss. Despite of the significant progresses, existing approaches only consider one checkpoint into influence estimation, neglecting the evolving dynamics of training, which has been shown to provide important clues for scoring each data influence [12, 23]. Specifically, the model behavior on each instance may vary during the progress of training via stochastic gradient descent, resulting in the importance scores of each sample to fluctuate throughout the optimization path of the deep networks. As illustrated in Figure 1, the subsets selected at different training checkpoints clearly contain various samples. Therefore, relying on static information from a single checkpoint for sample selection can lead to suboptimal subset construction and potential overfitting.

To address aforementioned problems, this work focuses on exploring a dynamic influence based data selection strategy. To do

so, we leverage the gradient-based dynamic influence estimation [23] to trace per-sample contribution across the model training procedure. Our theoretical analysis reveals that, for each checkpoint, selecting instance of higher projected gradient along the accumulated optimization direction leads to a better test performance. Specifically, given the unavailability of ground truth labels in AL, we rely on Kullback-Leibler (KL) divergence given by model outputs between checkpoints to measure gradient projection, eliminating the introduction of biased predicted label. Moreover, to extend our theoretical findings to a broader range of training progress, a straightforward approach is to save checkpoints and then calculate the gradient at each checkpoint [23]. Since expensive cost required for tracing massive unlabeled data in the AL setting, we propose to introduce an additional loss term to integrate the dynamics throughout the training progress, thus enhancing selection generalization. By incorporating these designs, we propose an universal AL framework, named Dynamic Influence Scoring for Active Learning DISAL, illustrated in Figure 2.

Extensive experiments on different tasks proves the validity and task-agnostic nature of our framework, including balanced and imbalanced image classification, semantic segmentation. We prove that our AL framework shows superior performances in comparison with other state-of-the-art baselines. Notably, our framework DISAL, utilizing 40% annotation budget on Caltech101 dataset [7], reaches a 93.58% accuracy, exceeding the model training based on the full training set at 92.59%.

We summarize our main contributions:

- We present a theoretical analysis demonstrating that the projected gradients on the accumulated optimization direction contribute to a valid influence scoring of unlabeled samples, which naturally leads to a criterion for data selection.
- Based on the theory, we introduce the DISAL to dynamically evaluate sample influence and includes effective dynamic loss term to trace model training progress for mitigating large-scale computation and memory cost.
- Comprehensive experiments show that DISAL significantly outperforms existing AL methods in a wide range of tasks and settings.
- Additionally, DISAL doesn't introduce any domain-specific design or extra learnable models, it is interpretable, flexible, and easy to implement.

## 2 RELATED WORK

### 2.1 Active Learning

The key objective of AL is to construct a data subset through designing an effective AL strategies, including mainstream feature distribution-based approaches and model uncertainty-based approaches.

As to the former, numerous studies have introduced k-means clustering and its variants to overcome data redundancy [2, 11]. Moreover, CoreSet [27] focuses on selecting a subset from an unlabeled data set that represents the whole set well. DAL [10] conceptualizes AL as a binary classification problem, labeling undistinguishable examples. VAAL [29] introduces an adversarial network to discriminate between labeled and unlabeled samples within the latent space encoded by the VAE [15].

As for uncertainty-based AL approaches, uncertainty has been widely used in AL to estimate samples' importance. Contrast to traditional uncertainty-based methods directly using the posterior probability [22, 22, 28, 34], DBAL [9] leverages Bayesian deep learning within AL, using Dropout for computing model uncertainty. Tran et al. [31] calculate the gradient with respect to the final layer of the deep network, prioritizing the annotation of diverse unlabeled samples with higher gradient norms. LL4AL [37] offers a more direct strategy by integrating an additional network to precisely predict each sample's loss. Furthermore, generative methods [6, 31, 41] have also been adopted in AL to generate samples with high uncertainty.

While existing two main categories of AL methods define the state-of-the-art baselines, they do not directly show how selected data impacts model performance. Several influence estimation-based methods aims to address this problem Liu et al. [21], Wang et al. [35]. Specifically, Liu et al. [21] prioritize the selection of unlabeled samples that can provide more positive influence, calculated by estimating its expected gradient. Similarly, drawing inspiration from Koh and Liang [16], Wang et al. [35] explore such an impact by offering theoretical evidence that the selection of unlabeled data with a higher gradient norm result in better test performance.

### 2.2 Influence Estimation Methods

Influence estimation is designed to assess the impact of specific training instances on a model's prediction for a given input. For models trained with gradient descent, the influence of training data is exerted solely through training gradients. Static gradient-based influence estimators determine influence by referencing only the final model parameters. A notable example is the influence function presented in [16], which define influence as the change in test loss caused by a minor perturbation to a training instance loss's weight. [25] extend this application of the influence function to assess unlabeled samples in semi-supervised learning scenarios. These static gradient-based influence estimation methods, however, only consider a single point in the procedure of training to estimate training instance's possible effect. Dynamic methods, however, provide a more comprehensive view by capturing the entire "story" of an instance's influence. TracIn [23] calculates influence by "tracing" gradient descent, aggregating changes in test loss each time training instance's gradient updates parameters.

Despite these advancements, the context of influence estimation in the setting of AL is entirely distinct. As far as we know, we are the first to introduce dynamic influence estimation to select unlabeled samples.

## 3 METHODOLOGY

Despite influence function-based methods establish a connection between sample contribution and model training loss, the single checkpoint-based selection hinders the construction of a subset with strong generalization. In this section, inspired by [23], we propose the Dynamic Influence Scoring for Active Learning (DISAL) framework as shown in Figure 2. In Section 3.1, We begin with the problem preliminary of AL. Following this, in Section 3.2, we provide a detailed theoretical derivation to evaluate the unlabeled

sample at checkpoints by projecting gradient on accumulated optimization direction. Section 3.3 extends the contribution evaluation to the entire training procedure, so as to provide a more comprehensive dynamic influence scoring. Finally, in Section 3.4, we show our proposed AL framework, DISAL.

## 3.1 Problem Preliminary

We define the AL procedure within the framework of standard supervised learning as follows: consider $\mathbf{X}$ to represent the input, and $\mathbf{Y}$ denote the corresponding output. Let $D = \{\mathbf{X}, \mathbf{Y}\}$ be the large pool of unlabeled data. A labeled training dataset is denoted as $D^L = \{(x_{1,1}), (x_2, y_2), ..., (x_n, y_n)\}$. We denote $D^U \subseteq D$ as the subset of unlabeled datasets. Throughout the AL procedure, a fixed number of samples are iteratively chosen from the unlabeled pool according to the acquisition strategy until reaching the sampling budget. In our method, we leverage the dynamic influence scoring as the criterion to select the most impactful instances on model performance. After adding the newly-annotated samples to labeled set, the optimization goal of neural network is to find the parameters $\theta$ that minimizes the training loss. Here we take the cross-entropy (CE) loss applied to image classification as an example, our goal is to train a label-efficient deep model $f(\cdot; \theta)$ where $\theta$ represents the parameters of the deep model to be optimized. Here the loss function is:

$$\mathcal{L}_{main}(\theta) = \frac{1}{n} \sum_{i=1}^{n} \text{CE} \left[ f_\theta(x_i), y_i \right]. \tag{1}$$

The ultimate objective of AL is to identify and annotate the most informative data, thereby enhancing the performance of the model.

## 3.2 Influence Scoring via Checkpoints by Projected Gradient

To calculate the dynamic influence score, we firstly explore how the unlabeled instance impact the model performance at checkpoints and propose a detailed theoretical derivation to evaluate the reduction in test loss.

We firstly assume that the model is updated via vanilla stochastic gradient descent, based on an idealized definition. Each training minibatch, denoted as $B(t)$, consists of a single instance, and gradient updates occur without momentum [26]. We first evaluate the influence of incorporating a single sample $x_n$ to the training set on the change in test training loss. Each iteration $t$ affects only the model parameters, updating the parameter vector from $\theta_t$ to $\theta_{t+1}$ using the training example $x_n$. In this context, the idealized influence of the training example $x_n$ on the test example $x'$ is denoted as the total reduction in loss:

$$I_{ideal}(x_n, x') = \ell \left( f_{\theta_t}(x'), y' \right) - \ell \left( f_{\theta_{t+1}}(x'), y' \right). \tag{2}$$

Although the definition of ideal influence has a strong theoretical motivation, it relies on single-instance batch setting and vanilla stochastic gradient descent, failing to work in practical scenarios. In fact, models are trained on batches containing hundreds or more of instances to ensure feasible training times. According to [23], to handle minibatches of size $b \geq 1$, a first-order Taylor approximation is used to approximate the impact of each training instance within a minibatch $B(t)$.

Given the typically small step sizes in parameter updates during training, the loss change for a test example within a given iteration can be estimated using a simple first-order Taylor expansion:

$$\ell \left( f_{\theta_{t+1}}(x'), y' \right) = \ell \left( f_{\theta_t}(x'), y' \right) + \nabla \ell \left( f_{\theta_t}(x'), y' \right) \cdot (\theta_{t+1} - \theta_t)$$
$$+ O \left( \|\theta_{t+1} - \theta_t\|^2 \right). \tag{3}$$

Here, the gradient $\nabla \ell \left( f_{\theta_t}(x'), y' \right)$ is calculated with respect the model parameters $\theta_t$. While stochastic gradient descent is utilized in training the model from $\theta_t$ to $\theta_{t+1}$, the formula for the change in model parameters is given by

$$\theta_{t+1} - \theta_t = -\frac{1}{b} \sum_{x_n \in B_t} \eta_t \nabla \ell \left( f_{\theta_t}(x_n), y_n \right) \tag{4}$$

where $\eta_t$ represents the step size at iteration t. Ignoring the higher-order term, which is on the order of $O(\eta_t^2)$, we derive a first-order approximation for the change in loss:

$$\ell \left( f_{\theta_t}(x'), y' \right) - \ell \left( f_{\theta_{t+1}}(x'), y' \right)$$
$$\approx \frac{1}{b} \sum_{x_n \in B_t} \eta_t \nabla \ell \left( f_{\theta_t}(x'), y' \right) \cdot \nabla \ell \left( f_{\theta_t}(x_n), y_n \right). \tag{5}$$

Consequently, for each training data $x_n$ within the minibatch $B_t$, we attribute the influence of $x_n$ on the test point $x'$ as the portion $\frac{1}{b} \eta_t \nabla \ell \left( f_{\theta_t}(x'), y' \right) \cdot \nabla \ell \left( f_{\theta_t}(x_n), y_n \right)$, effectively quantifying the contribution of each training instance with a first-order approximation.

However, the inherent randomness in training processes can negatively affect the above inference of sample influence. For instance, although intuitively identical training samples should receive the same influence score, their inconsistent appearance in the same minibatch can result in varying scores. We overcome it by evaluating the influence of each instance across the entire training data, not limiting the analysis to only the batches. In our context, during each iteration $t \in T$, all training instances are visited exactly once, updating the model parameter from $\theta_t$ to $\theta_{t+1}$. We assume a consistent step size $\eta_t$ between iterations. [23] provide a practical heuristic influence estimation at each iteration $t$:

$$I(x_n, x') = \eta_t \nabla \ell \left( f_{\theta_t}(x_n), y_n \right) \cdot \nabla \ell \left( f_{\theta_t}(x'), y' \right). \tag{6}$$

We assess example influence scoring on all the test sets. In AL settings, we assume the test set is $D^{test}$. For any instance $x_n$ involved in model training at iteration $t$, the influence on the test loss $I(x_n, D^{test})$ can be computed as:

$$I(x_n, D^{test}) = \eta_t \nabla \ell \left( f_{\theta_t}(x_n), y_n \right) \cdot \sum_{x' \in D^{test}} \nabla \ell \left( f_{\theta_t}(x'), y' \right). \tag{7}$$

The influence score $I(x_n, D^{test})$ indicates each training sample's potential to change test model performance. A higher positive influence score suggests a greater ability to reduce test loss. In contrast, the examples with a negative influence score are likely to be discarded as they contribute to an increase in test loss. It's notable that the test instance in question does not necessarily need to be in the test set but refers to any example whose prediction is being explained [23]. Given that the test set $D^{test}$ is unavailable at training phase, we use all labeled samples to act as an alternative, representing the data distribution of current tasks. Consequently,

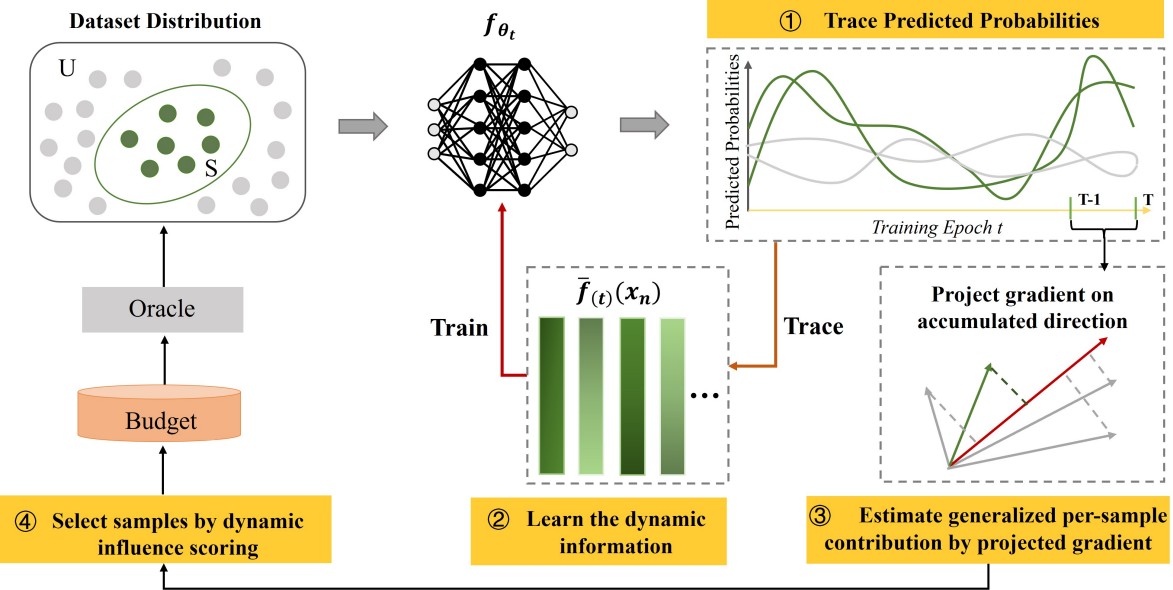

**Figure 2: The pipeline of our proposed DISAL framework. We first track prediction results throughout the training progress at each iteration. To integrate the dynamic information, the predictions $\bar{f}_{(t)}(x_n)$ are maintained and updated to augment the model training at each iteration. At the last iteration, the sample-wise gradient projection computing procedure follows to select samples with larger dynamic influence scoring and we utilize them to retrain the model.**

we define a practical scheme for influence estimation at iteration $t$ as follows:

$$I(x_n, D^L) = \eta_t \nabla \ell \left( f_{\theta_t}(x_n), y_n \right) \cdot \sum_{x' \in D^L} \nabla \ell \left( f_{\theta_t}(x'), y' \right). \quad (8)$$

Equation 8 indicates that per-sample contribution can be estimated by projecting its gradients onto the accumulated optimization gradient. Thus, we propose to utilize projected gradient to score sample influence at checkpoints. Compared with existing static gradient-based AL methods [35], which overlook optimization context, we contribute to a more valid contribution evaluation, which has been proved in the experiments in subsection 4.5.1.

However, Equation 8 involves calculating per-sample gradients over all training samples, which is not feasible due to computation and memory constraints. According to model optimization $\theta_{t+1} = \theta_t - \eta_t \sum_{x' \in D^L} \nabla \ell \left( f_{\theta_t}(x'), y' \right)$ at iteration $t$, the accumulated gradient is equal to $\frac{1}{\eta_t} (\theta_t - \theta_{t+1})$. Thus the influence described in Equation 8 is further reformulated in terms of the loss difference as follows:

$$I(x_n, D^L) = \frac{1}{\eta_t} (\theta_t - \theta_{t+1}) \eta_t \nabla \ell \left( f_{\theta_t}(x_n), y_n \right)$$
$$\approx \ell \left( f_{\theta_{t+1}}(x_n), y_n \right) - \ell \left( f_{\theta_t}(x_n), y_n \right), \quad (9)$$

which can be derived from the first-order Taylor expansion. Then the contribution of each sample to model performance can be assessed through changes in its loss. While $\ell$ is commonly represented by CE loss in the context of classification, the loss divergence can be defined as:

$$\ell \left( f_{\theta_{t+1}}(x_n), y_n \right) - \ell \left( f_{\theta_t}(x_n), y_n \right) = y_n^\top \cdot \log \frac{f_{\theta_{t+1}}(x_n)}{f_{\theta_t}(x_n)}. \quad (10)$$

Unfortunately, as discussed earlier, the one-hot encoding $y_n$ is unavailable for unlabeled data in the setting of AL. We overcome the challenge by replacing $y_n$ with predicted probability $f_{\theta_t}(x_n)$. So KL divergence is employed to capture a more comprehensive sample contribution information:

$$I(x_n, D^L) \approx f_{\theta_t}(x_n)^\top \cdot \log \frac{f_{\theta_{t+1}}(x_n)}{f_{\theta_t}(x_n)}. \quad (11)$$

In our experiments (refer to 4.5.2), we have observed that KL divergence outperforms the CE loss difference. We hypothesize that this superiority stems from the fact that the one-hot encoding of $y_n$ neglects non-target probabilities, thereby causing a substantial loss of information.

### 3.3 Dynamic Influence Scoring by Dynamic Loss

Since that exploring long-range training progress contributes to a more comprehensive sample evaluation, in this section, we simply extend our proposed influence scoring for each checkpoint in Section 3.2 to a dynamic influence scoring by introducing an additional loss. By capturing the entire "story" of influence, well-generalized samples are more likely to be identified.

To scale to long training processes, [23] replay the procedure by storing checkpoints at regular intervals, and sums up the influence throughout the model optimization. However, applying this approach to large-scale unlabeled data is computationally impractical, as it requires model inference on all the unlabeled data at each

checkpoint. To address this, we devise an efficient method by directly plugging the dynamic information into model training. Thus by the end of model training, more comprehensive sample-wise gradient projection would be obtained to select samples at the last iteration, via incorporating the entire of training progress.

Specifically, we develop a dynamic loss term, trained with dynamic prediction probabilities of labeled data, which is available during the training process. For simplicity, we construct the dynamic predicted probabilities $\overline{f}_{(t)}(x_n)$ by averaging the predicted outputs during $t$ iterations [30]. Then the dynamic predictions is updated at $t$ iteration as follows:

$$
\begin{aligned}
\overline{f}_{(t)}(x_n) &= \sum_{i=1}^{t} f_{\theta_i}(x_n)/t \\
&= \overline{f}_{(t-1)}(x_n) \cdot \frac{t-1}{t} + f_{\theta_i}(x_n) \cdot \frac{1}{t}.
\end{aligned} \tag{12}
$$

To learn the dynamic information effectively, the dynamic loss term is optimized by minimizing the KL divergence loss between the actual prediction $f_{\theta_t}(x_n)$ and the dynamic predicted probability $\overline{f}_{(t)}(x_n)$ at iteration $t$:

$$
\mathcal{L}_{\text{dynamic}} = \mathcal{L}_{\text{KL}}\left(\overline{f}_{(t)}(x_n) \| f_{\theta_t}(x_n)\right) \tag{13}
$$

By integrating the main task loss in Equation 1 and dynamic loss, we minimize an overall objective function as follows:

$$
\mathcal{L}_{\text{overall}}(\theta) = \mathcal{L}_{main}(\theta) + \lambda \cdot \mathcal{L}_{dynamic}(\theta), \tag{14}
$$

where $\lambda$ is a trade-off weight to control the effect of dynamic loss terms. The setting of $\lambda$ is examined in Section 4.5.3.

Finally, after model training, a more generalized dynamic influence scoring of each sample can be obtained at the final iteration $T$, by which point the entirety of the training dynamics are considered for sample selection.

$$
I(x_n) = f_{\theta_{T-1}}(x_n)^\top \cdot \log \frac{f_{\theta_T}(x_n)}{f_{\theta_{T-1}}(x_n)} \tag{15}
$$

## 3.4 Proposed AL Framework

Based on theoretical insights, we have formulated a universal AL framework integrating the dynamic influence scoring discussed in the preceding section. The framework begins with the selection of randomly annotated data to initiate training of the task model. Subsequently, after each training cycle, dynamic influence scoring is applied to select and annotate unlabeled data, which are used for retraining the task model. Notably, our framework introduces only minimal adjustments to the conventional AL workflow, ensuring its straightforward applicability across diverse scenarios. For a comprehensive understanding of our framework, please refer to Algorithm 1 and the Figure 2.

## 4 EXPERIMENTS

Considering the sensitivity of AL sampling algorithms to task-specific scenarios, we experimentally verify the effectiveness of our framework on two widely recognized visual tasks: image classification and semantic segmentation, demonstrating its broad applicability. Additionally, we explored its performance in class-imbalanced settings by using modified versions of the Cifar10 dataset. These

---

**Algorithm 1** DISAL

**Input**: Initial labeled dataset $D^L$; Unlabeled dataset $D^U$; AL cycles $K$; Iterations $T$; Dynamic influence scoring $I(\cdot)$.

1: **for** $k = 1, \cdots, K$ **do**
2:     **for** $t = 1, \cdots, T$ **do**
3:         Record predicted probabilities $f_{\theta_t}(x_n), x_n \in D^L$
4:         Update dynamic predicted probabilities $\overline{f}_{(t)}(x_n)$
5:                            ▷ *Defined in Equation 12*
6:         Train model integrating main task loss with dynamic loss
7:                            ▷ *Defined in Equation 14*
8:         **if** $t = T$ **then**
9:             Calculate dynamic influence scores $I(x_n)$
10:                            ▷ *Defined in Equation 15*
11:         **end if**
12:     **end for**
13:     $Q_t \leftarrow$ Sample queries $Q_t$ by scores
14:     $\overline{Q}_t \leftarrow$ Label queries $Q_t$ by annotator
15:     $D^L \leftarrow D^L \cup \overline{Q}_t$
16:     $D^U \leftarrow D^U \cup \overline{Q}_t$
17: **end for**

---

experiment results are reported across three trials, each employing different initial network weights and labeled data pool. Finally, the ablation study is conducted to provide deeper insights.

## 4.1 Image Classification

**Dataset.** In our experiments, we employ four image classification datasets: Cifar10 [17], Cifar100 [17], SVHN [22] and Caltech101 [7], with varying size and number of categories. Cifar10 and Cifar100 include 50,000 training and 10,000 testing images across 10 and 100 categories, respectively, with a resolution of $32 \times 32$ pixels. SVHN, similar in class count to Cifar10, includes 73,257 training and 26,032 testing images, without utilizing additional training data for fair comparison with other methods. Caltech101 comprises 9,146 images ($300 \times 200$ pixels) across 101 semantic categories, split into 8,046 for training and 1,098 for testing.

**Baselines and implementation details.** We compare DISAL with state-of-the-art AL baselines, including Dropout [8], Learning Loss [37], CoreSet [27], VAAL [29], CoreGCN [37], Boosting [35], TOD [14] and TiDAL [19]. In addition, the baseline methods also include the random sampling ("Random") and the model training based on the full training set ("Full Training").

Following conventional practices in AL [29], we employ ResNet-18 [13] as the backbone network for the primary task learner. Ninety percent of the images are allocated for training, while the remaining ten percent are reserved for testing purposes. The model is trained using the SGD optimizer, incorporating a momentum of 0.9 and a weight decay of $5 \times 10^{-4}$. For Cifar10 and Cifar100, the models undergo 200 epochs of training with a batch size of 128 and an initial learning rate of 0.1. Conversely, Caltech101 is trained for 50 epochs with a batch size of 64 and an initial learning rate of 0.01. Input images are standardized to dimensions of $224 \times 224$ pixels, accompanied by preprocessing techniques including random resizing, cropping, and horizontal flipping. For further elaboration, please refer to the supplementary material.

**Active learning setting.** For fair comparison, We follow the same AL setting and practice with baseline methods [29, 35]. In all tasks, we conduct 7 cycles of data annotation, expanding from 10% to 40% in 5% increments to cover various annotation budgets. In the initial cycle, we populate the labeled pool $D_0^L$ by randomly selecting

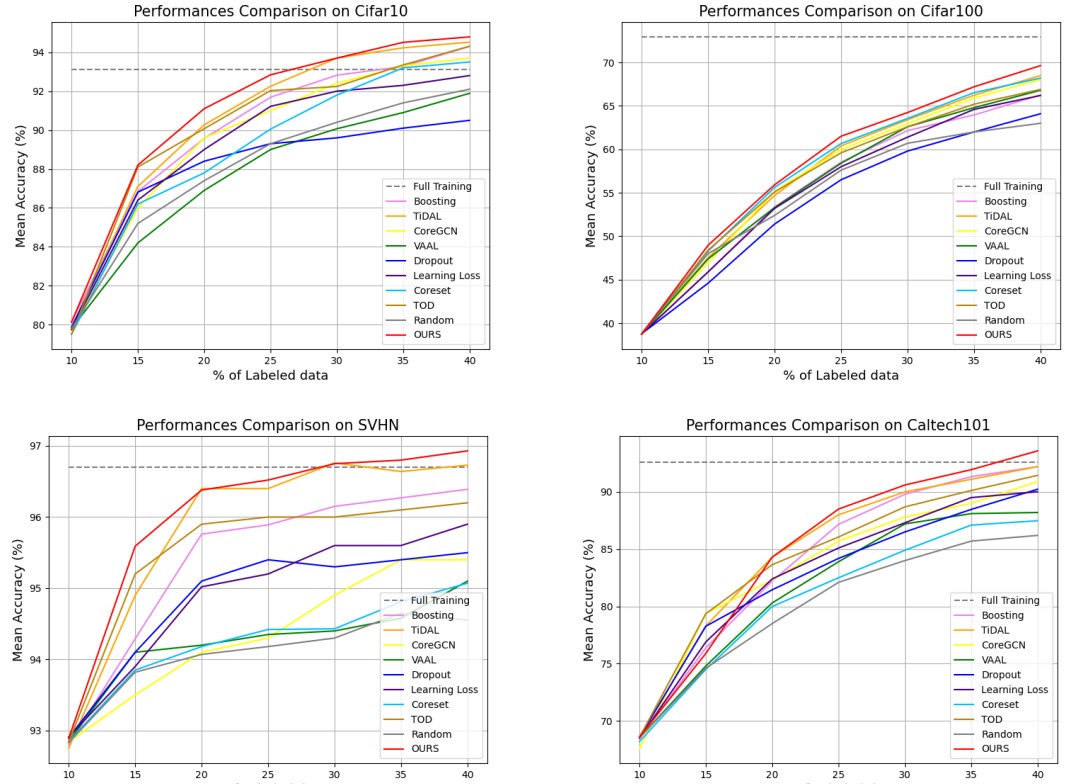

**Figure 3: Mean accuracy of different AL approaches on Cifar10, Cifar100, SVHN, Caltech101.**

10% instances for annotation from the unlabeled dataset to train an initial task learner. In each subsequent AL cycle, 5% of the data is selected from the unlabeled data pool $D^U$ for model retraining, according to the specific sampling strategy of each method. This process is repeated until the labeled portion of the dataset reaches 40%.

**Results.** Figure 3 illustrates the performance in terms of average test accuracies across different AL methods. Our approach, DISAL consistently outperforms baseline methods on all the datasets with a significant margin. Additionally, we make several observations that highlight the effectiveness of our proposed DISAL. First, DISAL achieves higher classification accuracies compared to other methods at various cycles, indicating that given different annotation budget such as 20%, our proposed DISAL still works best. Second, DISAL stands out in handling complex datasets like Cifar100 and Caltech101, which present greater challenges due to their larger number of classes. The superior performance and smoother curves in our method demonstrate its robustness. Remarkably, with 20,000 labeled instances, DISAL reaches a 69.64% accuracy in the final iteration, exceeding the next at 68.49%. Third, our model, when trained on 40% of the data using DISAL, outperform the "Full Training" on datasets including Cifar10, SVHN and Caltech101, e.g., 94.79% vs. 93.12% on Cifar10. It's also worth noticing that DISAL is the only method to achieve higher test accuracy than "Full Training" results on challenging Caltech101 benchmark, despite using only 40% labeled training data. Last, we also compare DISAL with semi-supervised method, TOD, which introduces a semi-supervised loss

to learn from unlabeled pool. Surprisingly, our proposed DISAL surpasses TOD in all datasets, even though TOD leverages additional unlabeled information and more training resources.

## 4.2 Semantic Segmentation

**Dataset and active learning setting.** To prove the effectiveness of AL methods in complex, large-scale environment, we evaluate our method on the widely recognized semantic segmentation benchmark dataset, Cityscapes [5]. Semantic segmentation task involves pixel-level classification, thus incurring significantly higher annotation costs. The Cityscapes dataset is a large-scale driving video dataset, consisting of video sequences of urban street scenes from 50 cities. The dataset includes 2,975 training and 500 validation images. For fair comparison, only the standard training and validation sets are utilized. In each AL iteration, 10% images are selectively sampled for the labeled training set until reaching the budget of 40%.

**Baselines and implementation details.** We choose the state-of-the-art AL methods for this experiment: Dropout [8], Learning Loss [37], CoreSet [27], VAAL [29], CoreGCN [37], Boosting [35] and TiDAL [19]. Following the widely used settings in [29], we adopt the 22-layer dilated residual network (DRN-D-22) [38] as the task model. The performance of each method was evaluated using the mean Intersection-over-Union (mIoU) metric. We crop the input images to a resolution of 688 × 688 pixels without the application of data augmentation. The dataset contains 19 categories. Detailed

**Table 1: Time (seconds) required for one AL cycle using an NVIDIA GTX 3090Ti GPU.**

|  | DISAL(Ours) | Random | TIDAL | Learning Loss | VAAL | CoreGCN | Coreset | Boosting |
|---|---|---|---|---|---|---|---|---|
| Cifar10 | **359.0** | 299.0 | 369.5 | 340.8 | 3737.4 | 457.6 | 445.5 | 376.5 |
| Cifar100 | **362.2** | 307.1 | 373.8 | 345.9 | 3683.7 | 451.2 | 421.9 | 387.2 |

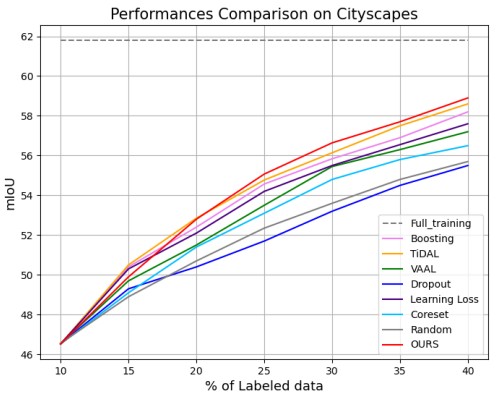

**Figure 4: Mean accuracy of different AL approaches on Cityscapes.**

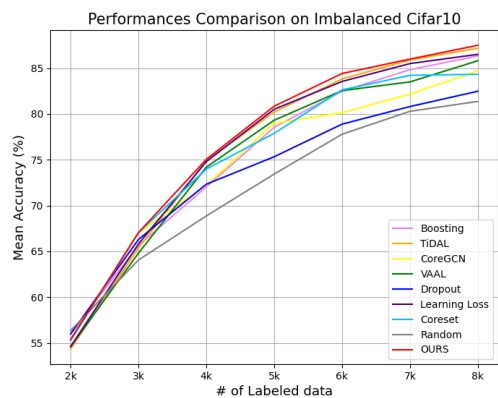

**Figure 5: Mean accuracy of different AL approaches on imbalanced Cifar10.**

information regarding other hyper-parameters is available in the supplementary material.

**Results.** The superior results in Figure 4 demonstrates the competence of DISAL in the challenging semantic segmentation task measured by mIoU. It is important to note that Cityscapes images are significantly larger than those used in classification benchmarks and require pixel-level classification, which indicates our method's robustness to complex data. Furthermore, DISAL's independence from task-specific knowledge eliminates the need for adjusting the training framework with changes in tasks, making it universally applicable. However, we observe that DISAL exhibits slightly lower performance than other premier methods in the first data selection. This discrepancy is likely due to the inherent instability of model training on challenging tasks with extremely limited data, where learning the dynamic information could potentially impact the model training.

### 4.3 Image Classification on an Imbalanced Dataset

**Dataset and Implementation Details.** To evaluate DISAL's performance in a more class-imbalanced context, we use the imbalance ratio (IR) of 10 to recompose the Cifar10 dataset. The distribution of images per class is as follows: airplane-500, automobile-1,000, bird-1,500, cat-2,000, deer-2,500, dog-3,000, frog-3,500, horse-4,000, ship-4,500 and truck-5,000. The experiments are conducted over the same seven AL cycles, varying from $2k$ of the labeled pool to $8k$. All additional experimental details are consistent with those for the balanced Cifar10, described in Section 4.1.

**Results.** Figure 5 shows the performances between DISAL and other baselines on the synthetically imbalanced Cifar10. Under the imbalanced settings, our proposed DISAL still clearly outperform all the baselines. Additionally, our findings reveal intriguing insights: Contrary to results from experiments on balanced Cifar10 and Cifar100 datasets, data distribution-based AL methods, such as

CoreGCN, perform less effectively than uncertainty-based methods, like Learning Loss in imbalanced settings. These findings suggest that distribution-based strategies suffer more in class-imbalanced settings compared to uncertainty-based approaches. Despite these challenges, DISAL shows consistent superiority across both class-imbalanced and balanced scenarios, demonstrating its versatility.

### 4.4 Computational Overheads

As described in Section 3.3, the extra computation for DISAL, apart from the task model training with an additional loss, is calculating the KL divergence of models outputs between checkpoints at the last iteration for selecting data. To compare computational overheads fairly, we assess one cycle duration of AL, which contains model training and a single-pass data selection, on Cifar10 and Cifar100 using various methods. As illustrated in Table 1, while DISAL incurs no additional memory costs such as saved checkpoints for dynamic influence estimation, its computational overheads are also comparable to those of other approaches under the same experimental settings. We attribute this advantage to its simple design, which relies solely on KL loss, without the requirement for additional model [19, 29] or costly clustering [27, 37].

### 4.5 Ablation Study

*4.5.1 Analysis on sampling strategy and dynamic loss.* To further understand the remarkable performance of DISAL, ablation studies are conducted to evaluate the contribution of its two main designs, influence scoring via checkpoints and dynamic loss.

As can be seen from the Table 2, our components notably enhance performance and robustness. For "DISAL w/o D", a variant that operates without dynamic loss, achieves comparable performance compared with other baselines, demonstrating the high effectiveness of influence scoring via checkpoints. And it's worth noting that on all the datasets, "DISAL w/o D" achieve consistent better performance than Boosting. Since Boosting doesn't take accumulated

**Table 2: Performance comparison with different AL approaches. "DISAL w/o S": DISAL without sampling strategy. "DISAL w/o D": DISAL without dynamic loss (Red: Best).**

| Datasets | VAAL | TIDAL | Core-set | Boosting | LL4AL | Random | DISAL-wo-S | DISAL-wo-D | DISAL(Ours) |
|----------|------|-------|----------|----------|-------|--------|------------|------------|-------------|
| Cifar10 | 91.9 | 94.5 | 93.5 | 92.4 | 92.8 | 92.1 | 92.2 | 94.3 | **94.8** |
| Cifar100 | 66.8 | 68.2 | 65.0 | 66.7 | 66.2 | 63.0 | 68.1 | 66.9 | **69.4** |
| Caltech101 | 88.2 | 92.2 | 87.5 | 92.2 | 90.0 | 86.2 | 89.8 | 92.5 | **93.6** |

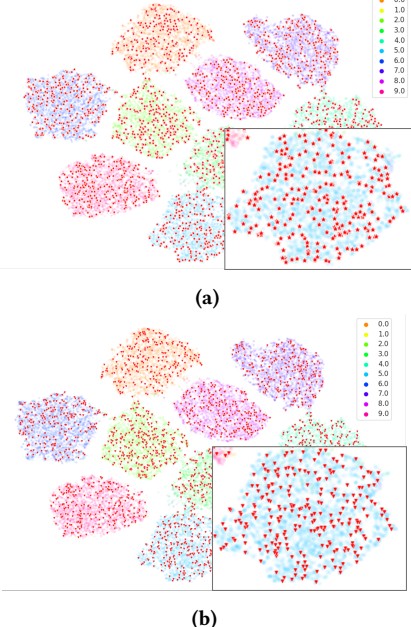

(a)

(b)

**Figure 6: T-SNE visualization of the Cifar10 dataset using (a) DISAL and (b) DISAL-wo-D (without dynamic loss). The red markers represent the data selected in the first cycle.**

optimization direction into consideration while calculate gradients, indicating that employing projected gradient could contribute to a more valid single samples influence contribution.

Despite comparable results, integrated with the dynamic loss, the accuracy of DISAL increases by 2.5% and 1.1% respectively against "DISAL w/o D" on Cifar100 and Caltech101. This highlights the need for the dynamic loss to trace training dynamics and evaluate more generalized sample contribution. To deepen our comprehension of dynamic influence scoring, we also show t-SNE [32] embeddings of the Cifar10 data, using DISAL and "DISAL w/o D". We use the model trained on these AL strategies to compute the representation of all unlabeled samples and those selected in the first AL cycle. Observations from Figure 6 reveal that both DISAL and "DISAL w/o D" strategies select samples that cover the entire data distribution of each class. However, selected samples from DISAL exhibit a broader distribution, particularly at class boundaries. As those samples on class boundaries are more uncertain with respect to the task model, they can significantly boost model training in the subsequent cycle. It is the reason why a substantial performance improvement has been observed while introducing the dynamic loss.

*4.5.2 Ablation on KL divergence loss.* As discussed in Section 3.2, we utilize the KL divergence loss to provide a more comprehensive assessment while the label of unlabeled data are unavailable. To

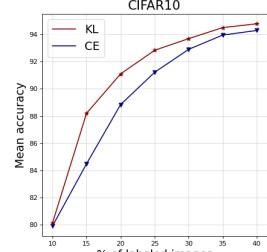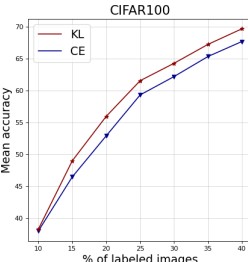

**Figure 7: KL and CE represent the data selection based on KL loss and CE loss difference, respectively.**

evaluate its effectiveness in characterizing sample contribution, we compare KL divergence loss with CE loss difference, which temporarily "observes" its label. As illustrated in Figure 7, although CE loss difference is blessed with labels which is unreasonable in AL, KL loss still exhibits an absolute advantage at all the cycles on Cifar10 and Cifar100, proving the effectiveness of our KL divergence loss.

**Table 3: Ablation studies on the performance of trade-off weight.**

| $\lambda$ | Number of Cifar100 Images | | | | | | |
|-----------|------|------|------|------|------|------|------|
| | 10% | 15% | 20% | 25% | 30% | 35% | 40% |
| 1 | **38.24** | **48.96** | **55.92** | **61.51** | **64.24** | **67.2** | **69.64** |
| 0.5 | 38.06 | 48.64 | 56.44 | 61.24 | 64.12 | 66.47 | 68.86 |
| 0.1 | 38.14 | 48.43 | 55.2 | 60.5 | 64.51 | 65.63 | 67.86 |

*4.5.3 Ablation on the effect of trade-off weight $\lambda$.* Further, we analyze the effect of trade-off weight $\lambda$ in Equation 15, with which we combine main task loss and dynamic loss for model training. The results are presented in Table 3. Notably, when $\lambda$ is equal to 1, our AL framework DISAL attains optimal results across various annotations.

## 5 CONCLUSION

This paper first presents a theoretical analysis demonstrating that selecting samples with higher projected gradients on the accumulated optimization direction throughout training can lead to enhanced performance. By integrating dynamic model training, we propose a task-agnostic framework called Dynamic Influence Scoring for Active Learning (DISAL), evaluating per-sample contribution throught the training procedure. Our extensive experiments across diverse tasks confirm the validity of our theoretical findings and showcase the effectiveness of the proposed framework. Looking ahead, our future work will delve into incorporating a learnable trade-off weight to more adeptly capture dynamic information, thereby advancing the field of AL even further.

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
