# OpenReview forum: "Tracing Training Progress: Dynamic Influence Based Selection for Active Learning"
_acmmm.org/ACMMM/2024/Conference — MM2024 Poster_

### Official Review · Reviewer_sL1v · 2024-05-05

**Rating:** 2
**Confidence:** 4

**Summary:**

The paper presents a novel active learning framework, "Dynamic Influence Scoring for Active Learning" (DISAL), which aims to enhance model performance by dynamically tracing the impact of unlabeled instances on model performance throughout the training process. The authors argue that classical active learning (AL) methods often rely on human expertise to design sampling strategies, which limits their scalability and generalizability. The proposed DISAL framework employs gradient-based dynamic influence estimation to trace per-sample contribution across the model training procedure. The theoretical analysis suggests that selecting instances with higher projected gradients along the accumulated optimization direction at each checkpoint leads to improved performance. The framework is tested across multiple tasks, demonstrating significant improvements over existing state-of-the-art AL methods.

**Strengths:**

The proposed framework is universal and task-agnostic, making it broadly applicable to various domains.

The paper presents extensive experiments across multiple datasets and tasks, demonstrating the efficacy of the DISAL framework.

**Limitations:**

1. **Complexity in Implementation:** The method’s reliance on dynamic gradient tracking and additional loss terms could complicate implementation and increase the computational overhead, despite claims of computational efficiency.
2. **Limited Discussion on Scalability:** While the method is shown to be effective on benchmark datasets, the paper lacks a detailed discussion on scalability, especially for very large datasets or in resource-constrained environments.
3. **Dependency on Hyperparameters:** The performance of DISAL seems to be sensitive to specific hyperparameters like the trade-off weight λ, but the method for selecting these parameters isn’t comprehensively discussed, which could affect reproducibility and practical deployment.

**Suitability:**

2

---

### Official Review · Reviewer_6GTY · 2024-05-19

**Rating:** 4
**Confidence:** 2

**Summary:**

The paper is about a task-agnostic framework DISAL, which integrates dynamic model training to evaluate per-sample contribution throughout the training procedure. The paper firstly analyzes that the unlabeled data's projected gradient on the accumulated training gradient can be interpreted as its influence score, then the framework aims to enhance performance by selecting samples with higher projected gradients on the accumulated optimization direction. Besides, an auxiliary dynamic loss is introduced to help trace the long term progress.

**Strengths:**

1. The derivation of the key equation 11 is interesting and reasonable.
2. The experiments are relatively comprehensive. Besides the image classification task, segmentation task is also included to verify the effectiveness. Imbalanced dataset which is more difficult is also studied.
3. According to the ablation studies, the two introduced method --- sampling strategy and dynamic loss, bring large bonus on the baseline accross different datasets.
4. While bringing improvements on the accuracy, the proposed DISAL still maintains a relatively fast inference speed.

**Limitations:**

1. Some expressions in the paper seem to be confusing and need further refinement or illustration. For example, what does the statement "For instance, although intuitively identical training samples should receive the same influence score, their inconsistent appearance in the same minibatch can result in varying scores." mean?
2. Although the ablation study shows the introduced dynamic loss helps, it is somehow confusing. The alignment between past progress and the prediction at present does bring the dynamic information, but the future progress would also be restricted. This may occur when more fluctuating training like training on the imbalanced datasets or other modal data.
3. Although the author state that the method is task-agnostic and can be generalized, it is only tested on image modal. The verification on other modalities is highly recommended and encouraged.

**Suitability:**

2

---

### Official Review · Reviewer_ZrNP · 2024-05-23

**Rating:** 3
**Confidence:** 2

**Summary:**

Authors introduce the DISAL to dynamically evaluate sample influence and includes effective dynamic loss term to trace model training progress. Authors projected gradients on the accumulated optimization direction contribute to a valid influence scoring of unlabeled samples.

**Strengths:**

[1] The method proposed in this paper is simple and effective.
[2] This paper is clearly laid out.

**Limitations:**

[Major] Whether this method truly fits within the active learning framework, as AL may require a labeling process after the selection and before further training. This method may not be applied at all in practice.
[Minor] I acknowledge that the authors have indeed made novel contributions: 1) proposing to calculate gradient projections at multiple checkpoints instead of a single point to capture training dynamics; 2) introducing an additional dynamic loss term to more efficiently integrate training dynamics information. However, these contributions are largely built upon the existing theoretical proof that selecting unlabeled data with higher gradient norms leads to better test performance.

**Suitability:**

3

---

### Official Review · Reviewer_ZQ3g · 2024-05-24

**Rating:** 4
**Confidence:** 3

**Summary:**

The paper introduces a novel active learning method that dynamically tracks the training process to select samples based on the current model status. The authors review previous works and argue that optimal selection strategies can change throughout training. Theoretical analyses demonstrate that selecting samples with high projected gradients can be beneficial. Building on this, the authors propose DISAL, which dynamically evaluates the influence of unlabeled samples on the current model and successively selects the most important ones. Additionally, a dynamic loss term is introduced to capture global influence. Experiments show that DISAL outperforms competing methods in both classification and segmentation tasks.

**Strengths:**

1. The motivation behind the idea is both sound and clear, demonstrating the insight that optimal selection strategies can evolve throughout the course of training. The empirical results presented in Figure 1 convincingly support this concept.
2. Theoretical analyses in Section 3.2 offer a stronger theoretical guarantee for the proposed method compared to heuristic approaches. This robust theoretical foundation is particularly crucial for the reliability and efficacy of an active learning method.
3. The experimental evaluation is comprehensive, encompassing a variety of tasks, including both classification and segmentation. It also addresses both balanced and imbalanced settings, ensuring the method's versatility. The visualization results, such as those presented in Figure 6, are especially interesting and provide valuable insights that help to better understand and appreciate the method's effectiveness.
4. The organization of the paper is logical and coherent, with writing that is clear and fluent. This makes it easy for readers to understand the content and follow the arguments presented, enhancing the overall readability and impact of the paper.

**Limitations:**

1. The concept of "Dynamic Loss" introduced in Section 3.3 could use some additional clarification. For instance, what exactly does "capturing the entire 'story' of influence" mean? Additionally, it would be helpful to understand why minimizing the KL divergence in Equation 13 is beneficial. Could this process slow down training by constantly aligning with the previous predictions?
2. The experiments seem to indicate that DISAL is effective for unimodal tasks. However, is there any indication or speculation on whether it could extend its benefits to multimodal classification tasks?

**Suitability:**

2

---

### Meta-Review · Area_Chair_hfRD · 2024-07-02

**Recommendation:** Accept (Poster)
**Confidence:** 4

**Metareview:**

This work aims to improve active learning performance by dynamically evaluating the influence of unlabeled samples throughout the training process. The proposed method argues that dynamic selecting strategies are preferring during the training procedure. Gradient projections at multiple checkpoints are demonstrated to be good indicator for sample selection. A dynamic loss term is introduced to capture global influence. The proposed method outperforms competing methods in both classification and segmentation tasks.

Pros:
- The concept of dynamically evaluating sample influence and integrating a dynamic loss term is innovative. This approach provides a new perspective on active learning (AL) by focusing on the influence of samples during the training process​.

- This work includes extensive experiments across multiple datasets and tasks, demonstrating significant improvements over existing state-of-the-art AL methods.

Cons:
- The reliance on dynamic gradient tracking and additional loss terms might complicate implementation and increase computational overhead.

- The framework is mainly tested on image data, while the authors claim it to be task-agnostic. Additional evaluations on other modalities is recommended to confirm its generalizability. During the rebuttal, authors presented additional experiments on text classification tasks, which alleviated the above concerns.

Overall, this paper proposed novel ideas for active learning with solid empirical results on multiple tasks. Reviewers are generally positive after reading the rebuttal. Therefore, it is recommended for acceptance.